# Intriguing Properties of Modern GANs

## Abstract

Modern GANs achieve remarkable performance in terms of generating realistic and diverse samples. This has led many to believe that "GANs capture the training data manifold". In this work we show that this interpretation is wrong. We empirically show that the manifold learned by modern GANs does not fit the training distribution: specifically the manifold does not pass through the training examples and passes closer to out-of-distribution images than to in-distribution images. We also investigate the distribution over images implied by the prior over the latent codes and study whether modern GANs learn a density that approximates the training distribution. Surprisingly, we find that the learned density is very far from the data distribution and that GANs tend to assign higher density to out-of-distribution images. Finally, we demonstrate that the set of images used to train modern GANs are often not part of the typical set described by the GANs' distribution.

## 1 Introduction

Starting with their original formulation, Generative Adversarial Networks (GANs, Goodfellow et al. 2014) quickly gained enormous popularity. As their popularity increased, so did the number of approaches dedicated to evaluating the quality and fidelity of the images created by GANs as compared to the true distribution (e.g. Heusel et al. 2017; Richardson & Weiss 2018; Arora et al. 2018; Kynkäänniemi et al. 2019). These methods exposed many problems in the distributions of images generated by earlier versions of GANs, such as mode collapse (Goodfellow 2016 and artifacts within generated images (Karras et al. 2021). Such developments were carried out with the understanding that "the GAN's training criteria has a global minimum at the true distribution" (from Theorem 1 of Goodfellow et al. (2014)).

Following these evaluations, there was a refinement of GANs and the mentioned failures have become much less prominent. For example, FID (Heusel et al. 2017) and recall (Kynkäänniemi et al. 2019) both check whether the full diversity of the training distribution is captured by the generative model. GANs with low FID and high recall are deemed not to suffer from mode collapse, as they supposedly capture the full support of the training data. Due to their impressive performance in such metrics, many applications utilize GANs with the working assumption that GANs learn the "training data manifold" (Schlegl et al. 2017). This assumption has even carried through to domains where an incorrect model of the true distribution can incur heavy risks, such as super-resolving MRI scans (Bendel et al. 2022), simulations for self-driving cars (Zhang et al. 2018a), detecting cancer through GAN inversions (Schlegl et al. 2017), recovering frequencies in astrophysical images ( Schawinski et al. 2017) and more.

In this work we study GANs with excellent performance and question whether these GANs are actually a good approximation of the true data manifold. By critically analyzing GANs through this lens, we find that they behave in ways counter to popular belief.

As mentioned above, modern GANs are frequently thought of as manifold methods, wherein the GANs' generator captures the image manifold. Surprisingly, we find that training examples are not part of the learned manifold. We go on to demonstrate that the GAN manifold passes almost the same distance from the class of images the generator was trained on as other classes, even when measured with a perceptual distance (Zhang et al. 2018b). This failure is intriguing: a GAN trained on ships, for instance, is only able to generate images of ships and not other object classes, but cannot generate ships similar to those it has seen

during training. Furthermore, we find that the GAN manifold passes unusually close to images outside the training domain. In particular, we show that the GAN manifold passes closer to SVHN images than to the images it was trained on, even when distances are measured using a perceptual metric (Zhang et al. 2018b).

While GANs are most commonly viewed as manifold methods, we demonstrate how analyzing them only in these terms might not capture the full breadth of their capabilities. Specifically, the manifold view ignores the density the generator assigns to different parts of space and whether or not the manifold is more abundant in regions around the true distribution. To account for this, we analyze the density the GAN's generator assigns to samples in terms of the average test log-likelihood. While mostly overlooked in the GAN literature, the log-likelihood is widely used to evaluate other families of generative models, such as score based models (Song et al. 2020) and normalizing flows (Kingma & Dhariwal 2018). Moreover, the log-likelihood was used in the past to show that large families of generative models assign higher likelihood to images outside the training distribution than those used in training (Nalisnick et al. 2018) and that many conditional generative models are weak classifiers (Fetaya et al. 2019). However, works such as those by Nalisnick et al. (2018) focused on models where the likelihood is tractable, which typically have much worse FID scores than modern GANs. There is scarce literature of this vein concerning generative models that show excellent results on metrics other than average test log-likelihood.

By evaluating GANs with low FID as density estimators, we show that modern GANs under-perform in terms of average test log-likelihood when compared to models with much worse FID. We go on to show that GANs are biased towards images with larger "flat" areas. Indeed, the log-likelihood assigned by the GAN is anti-correlated with the amount of local variance in the image.

Finally, we question whether it is plausible that the images used to train the GAN were sampled according to the generative process of the GANs. We do this by testing whether the training images are part of the typical set of the GAN, a test previously used to determine whether a set of images are outliers according to a generative model (Nalisnick et al. 2019). We find that the training examples are *not* part of the typical set of the GAN; that is, the probability that the GAN would generate the training images is essentially zero.

To summarize, in this paper we discuss four intriguing properties of modern GANs:

- The learned manifold does not pass through the training examples.

- The learned manifold is closer to out-of-distribution images than to in-distribution images.

- The density model learned by by the GAN assigns higher density to out-of-distribution images.

- The training images are not in the typical set of the GAN.

## 2 Background

### 2.1 Evaluation of Generative Models

State of the art GANs (Sauer et al. (2022); Karras et al. (2020b); Kang et al. (2023)) are most often evaluated by methods that analyze the statistics of the generated samples (Heusel et al. 2017; Kynkäänniemi et al. 2019; Richardson & Weiss 2018); the numbers usually reported are IS, FID and precision/recall. Almost all of these approaches define a statistic that is zero only if the GAN-generated and true samples are from the same distribution. Methods such as these are extremely effective in determining whether the generative model has learned the correct statistic but are somewhat limited. One such limitation is that for data in very high dimensions, many samples might be needed to determine if the true and generated distribution are distinct. Another is that the difference between the distributions is only measured on particular statistics, ignoring the others.

Besides GANs, many generative models utilize the average test log-likelihood (LL) as an additional evaluation metric, since high test LL is equivalent to low KL divergence between the generative distribution and the true data distribution. For example, score-based models (Song et al. 2020) and normalizing flows (Kingma & Dhariwal 2018; Nalisnick et al. 2018; Fetaya et al. 2019) report the test LL or the negative test LL as a

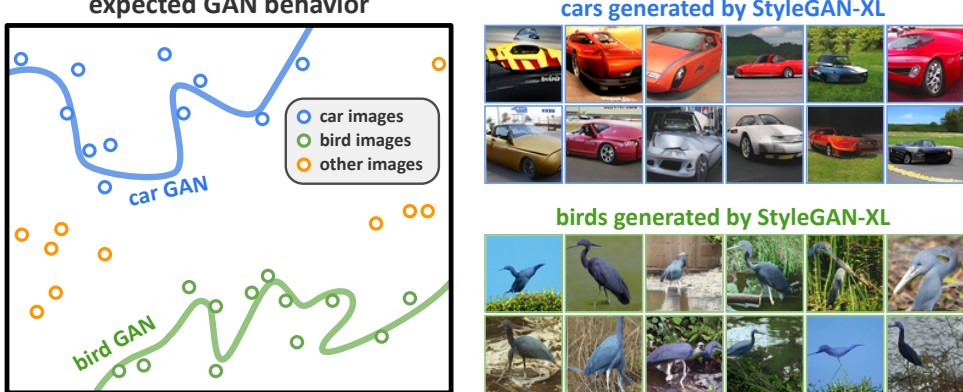

Figure 1: Modern GANs work amazingly well, to the point that it is expected that they capture the data manifold (left). It is typically assumed that because GANs can generate realistic images (right) that they capture the true data manifold. In this paper we show that this assumption is false.

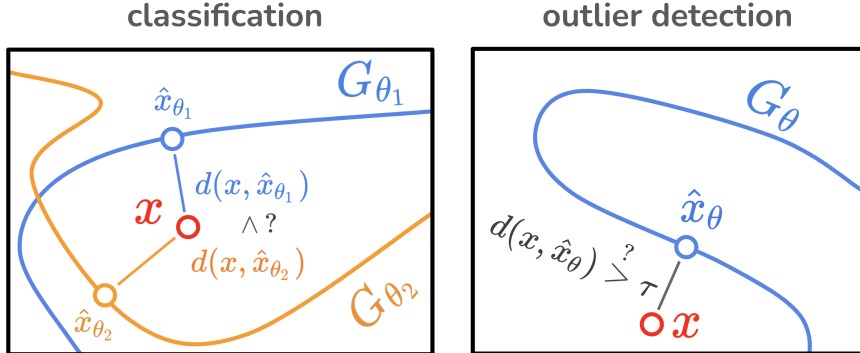

Figure 2: If GANs capture the true data manifold, it should be possible to use them for classification (left) and outlier detection (right). In both cases a sample $x$ is projected to the manifold, giving $\hat{x}_\theta$, and the distance between the original and the projection is used either to infer the class or whether the sample is an outlier.

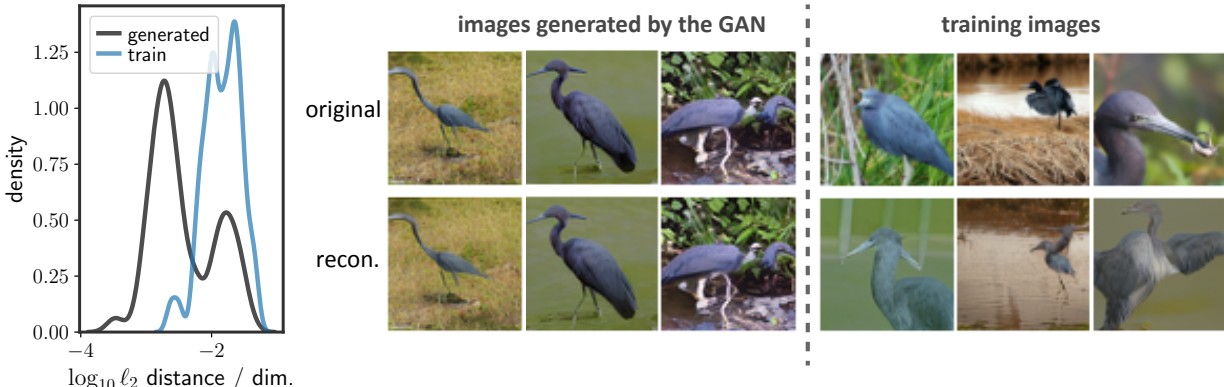

Figure 3: **Left:** $\ell_2$ distance (lower is better) of reconstructions of generated and train images under StyleGAN-XL (Sauer et al. 2022) trained on ImageNet (when only $z$ is optimized). The distribution of reconstruction errors for training images is distinct from the same for images generated by the GAN, implying that the training images are not part of the GAN manifold. **Right:** examples of projections to the GAN. Generated images are part of the manifold (and can be reconstructed), but training images are not. For training images, the reconstructed images are realistic birds but they are different birds.

way to compare the quality of different models. Since the LL is never explicitly used or calculated in GANs, it was largely ignored when comparing such models and calculated in only relatively few studies for small scale models on simple images, such as MNIST (Wu et al. 2016; Theis et al. 2015).

In the past, generative models were also evaluated through their performance in inference tasks, such as generative classification (e.g. Fetaya et al. 2019; Hinton et al. 1995). In generative classification, a generative model $p_{\theta_c}(x|c)$ is separately trained for each class $c$ in the data. An unseen sample is then classified by determining which class-conditional LL is the largest:

$$\hat{c}(x) = \arg\max_c \log p_{\theta_c}(x|c) \tag{1}$$

where $\hat{c}(x)$ is the predicted class for sample $x$. If $p_{\theta_c}(x|c) \approx p_{\text{data}}(x|c)$ for all classes $c$, then the above classification scheme is guaranteed to be optimal (Duda et al. 1973). On the flip side, if the classification performance is sub-optimal, then in no uncertain terms $p_{\theta_c}(x|c) \neq p_{\text{data}}(x|c)$. Using inference tasks to evaluate generative models can expose biases in the generative distribution, and is potentially a sample-efficient way to find out whether a generative model hasn't learned the true data distribution.

## 2.2 Annealed Importance Sampling

Calculating the LL of any decoder-based model involves calculating the following integral:

$$p_\theta(x) = \int p_\theta(x|z)p(z)dz \tag{2}$$

where $p(z)$ is the prior over the latent dimension and $p_\theta(x|z)$ is the probability of generating $x$, using the model, with the latent code $z$. Unfortunately, analytically solving this integral is intractable for complex decoder-based models. Instead, approximate methods such as Markov Chain Monte Carlo (MCMC) must be used in order to calculate the LL. In particular, Wu et al. (2016) showed that annealed importance sampling (AIS, Neal 2001) can be used to accurately approximate the LL of GANs.

AIS is an MCMC approach that uses multiple intermediate distributions in order to estimate the integral in equation 2. Wu et al. (2016) showed that an accurate estimate can be achieved, using many intermediate distributions (i.e. long MCMC chains) and multiple chains. When used to calculate the *log* of equation 2, the estimate is only a stochastic lower-bound (Grosse et al. 2015) of the LL that gets tighter as more intermediate distributions are introduced. For a more detailed explanation, see appendix A or Wu et al. (2016).

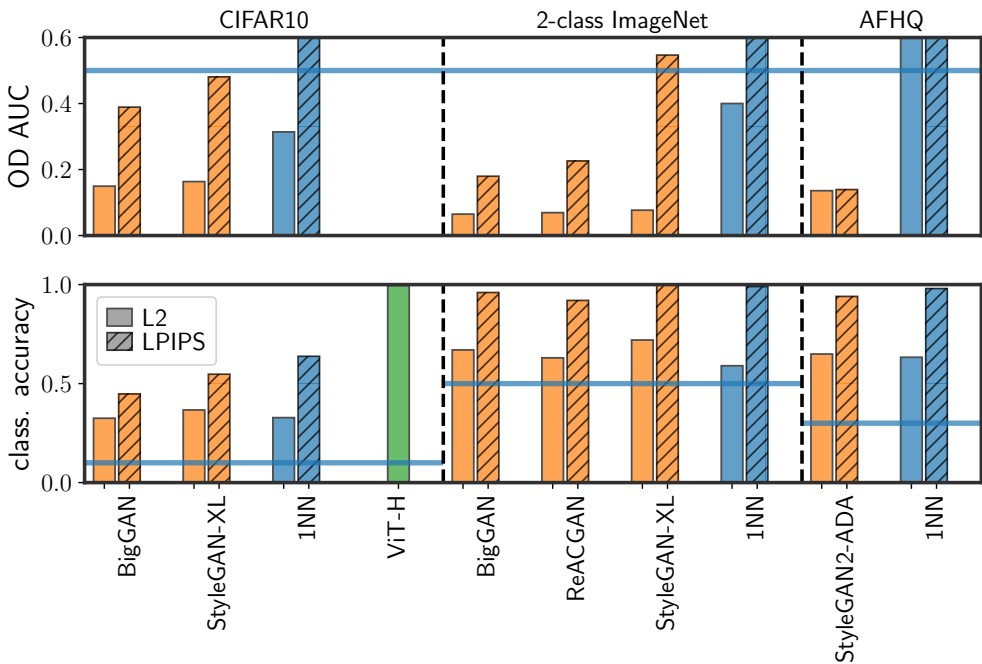

Figure 4: Performance of different GANs at the task of outlier detection (top) and classification (bottom). In all cases, both the $\ell_2$ (plain) and the LPIPS (hatched, Zhang et al. 2018b) distances are used. The GANs are compared to a 1-nearest neighbor (1NN) baseline. For CIFAR10, all methods are also compared with ViT-H (Dosovitskiy et al. 2020). The GANs always underperform, compared to the 1NN baseline.

## 3 Are GANs Good Manifold Methods?

GANs transform samples from a low dimensional latent space, $\mathcal{Z}$, into the larger dimension of the data $\mathcal{X}$, effectively describing a manifold. In much of the literature, it is believed that the GAN captures the true low dimensional behavior of the training data (e.g. Menon et al. 2020; Schlegl et al. 2017; Xiao et al. 2021; Bendel et al. 2022; Schawinski et al. 2017). A simple schematic of this assumption can be seen in figure 1: it is usually expected that the images from the training distribution are very close to, or exactly on, the learned manifold. Furthermore, points that are not part of the training distribution are expected to be far from the learned manifold, whether they are natural images or not.

If GANs truly behave as believed and shown in figure 1, it should be possible to use them for classification and outlier detection (Schlegl et al. 2017). For classification, an image is separately projected onto GANs trained on each of the classes and is classified according to the manifold whose projection was closest to the original image. A schematic of this can be seen in figure 2 (left). Outlier detection can be carried out analogously, where the point is projected to the GAN manifold and if it's distance from the original point is too large it is deemed an outlier, as shown in figure 2 (right).

### 3.1 Inference with the GAN Manifold

We test whether modern GANs can be considered as manifold methods using a range of GANs and various datasets. The GANs we use are: StyleGAN-XL (Sauer et al. 2022) with an FID of 1.94 on ImageNet and 1.88 on CIFAR10, ReACGAN (Kang et al. 2021; 2022) with an FID of 8.19 on ImageNet and 3.87 on CIFAR10, BigGAN-DiffAug (Zhao et al. 2020) with an FID of 8.7 on CIFAR10, BigGAN (Kang et al. 2022; Brock et al. 2018) with an FID of 8.54 on ImageNet, and StyleGAN2-ADA (Karras et al. 2020a) with an FID of 3.55 on AFHQCat and 7.4 on AFHQDog. To project the images on the GAN, we optimize the latent $z$ code from multiple restarts and choose the optimized code with the smallest reconstruction error. Further details can

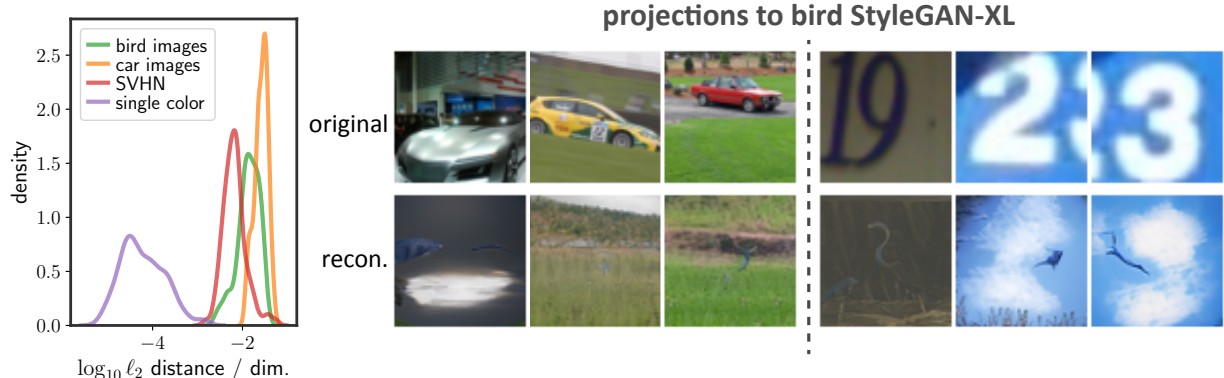

Figure 5: **Left:** $\ell_2$ reconstruction errors (lower is better) for the reconstruction of different image groups by StyleGAN-XL trained on birds. The reconstruction error for birds or cars is similar, while there exist images that the GAN can reconstruct much better than those it was trained on. **Right:** examples for StyleGAN-XL reconstructions of car images and rescaled SVHN images. In all cases, the GANs capture the overall image structure, but don't retain the identity of the main object in the image, leading to projections which are far from the image whether in the training domain or not.

be found in appendix B. All of the GANs chosen are able to generate highly realistic images and seemingly capture the full diversity of the datasets they were trained on.

In the task of classification, we analyze the performance of the GANs on a subset (1000 test samples) of the CIFAR10 dataset, a binary problem from ImageNet ("sports car" class versus the "blue heron" class) and a subset (1000 samples) of AFHQ images, where the classes are AFHQDog, AFHQCat and AFHQWild. For outlier detection in all datasets, we consider images of a single color and rescaled versions of SVHN images as outliers and report the area under the ROC curve (OD AUC).

The first assumption we question is whether the GAN manifold actually passes through the training examples. For a given input $x$ we can determine if it is on the GAN manifold by using gradient descent to approximately solve the optimization problem:

$$z^* = \arg\min_z \|G(z) - x\| \tag{3}$$

Surprisingly, it turns out that most of the training points are *not part of the GAN manifold*. Figure 3 (right) shows that for samples generated by the GAN, the optimization algorithm succeeds in finding a latent $z$ so that the generated images is almost identical to the input image $x$. But for images from the training set, the best $z$ found by the optimization algorithm generates an image that looks very different, although realistic. The figure also shows histograms of the reconstruction error of training samples found by inverting them from the $\mathcal{Z}$ space for StyleGAN-XL on ImageNet (for results on more GANs, see appendix C). As a baseline, samples generated by the GAN are also inverted and the histogram of their reconstruction error is shown as well. In general, the reconstruction error of training images is an order of magnitude worse than the reconstruction error of a generated image.

We then test the classification accuracy and outlier detection capabilities of the GANs. As mentioned, if the GANs have learned something close to the true data manifold, then they should act as good classifiers and outlier detectors. As a baseline, we compare the GANs to a simple 1-nearest-neighbor (1NN) estimator. Under the 1NN estimator, a sample is classified according to the class of its nearest neighbor. In outlier detection, a sample is considered an outlier if it is further than a pre-defined threshold from its nearest neighbor in the training distribution. This baseline is analogous to a GAN that has completely memorized the training examples and can't generate anything outside the training examples.

We again find that the learned manifolds do not behave as expected. Using the GAN's manifold for classification results in poor accuracy, as shown in figure 4 (bottom, in orange). The GANs are usually worse than

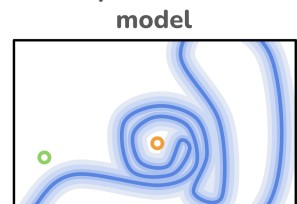

Figure 6: An example of a case where simply calculating the distance from the manifold might not tell the whole story. In this case, both $x_1$ and $x_2$ are the same distance from the manifold but the manifold is more abundant near $x_1$, making images in its region more plausible (left). By adding an observation model (right), queries to the model are transformed to calculating the log-likelihood under this observation model, essentially integrating over all possible areas of the manifold. With the observation model (right), $x_1$ is assigned a higher log-likelihood than $x_2$, even though they are equally distant from the manifold.

| Model | Dataset | LL (bits/dim.) ↑ | FID ↓ |
|---|---|---|---|
| BigGAN-DiffAug | CIFAR10 | -6.68 | 8.7 |
| StyleGAN-XL | CIFAR10 | -7.27 | 1.88 |
| Glow (Kingma & Dhariwal, 2018) | CIFAR10 | -3.35 | 3.35 |
| DDPM++ (Song et al., 2020) | CIFAR10 | -2.99 | 2.92 |
| BigGAN | ImageNet | -7.59 | 8.54 |
| StyleGAN-XL | ImageNet | -7.29 | 1.94 |
| Glow (Kingma & Dhariwal, 2018) | ImageNet | -3.81 | - |
| VDM (Kingma et al., 2021) | ImageNet | -3.4 | - |
| StyleGAN2-ADA | AFHQ | -7.75 | 3.55 |

Table 1: GAN performance as a density estimator next to other generative models. Although GANs sometimes come ahead in terms of FID, their average test log-likelihood is always lower.

the simple 1NN classifier (in blue), even when using a more perceptual distance such as LPIPS (Zhang et al. 2018b). The performance of GANs as outlier detectors is not much better, as shown in figure 4 (top) - the performance of most GANs is worse than random, even when using the perceptual distance. On the other hand, the 1NN outlier detector is always better than chance when using LPIPS.

We can better understand this poor performance by explicitly looking at the projections to the GAN manifold. Figure 5 (left) shows the reconstruction errors that different image groups have under a StyleGAN-XL trained on ImageNet images of blue herons (birds). Many images of sports cars have the same reconstruction error as images of birds, explaining why GANs are so bad at classifying between different images. Moreover, the GAN is able to reconstruct images of a single color and at times images from SVHN better than images from the distribution the GAN was trained on. Figure 5 (right) shows projections of images of sports cars (top) and SVHN (bottom) onto the bird GAN. The projection of any image keeps the overall image structure, even for images from the wrong class. Because of this, the projection's distance from the original image is about the same whether the original was from the training distribution or not.

Such behavior is surprising. Images of sports cars and SVHN digits are very different than those of birds, but the GAN is still able to reconstruct much of the context of the images. In images of cars, the GAN reconstruct the background, removing the car. For SVHN, the dominant colors and positions of the digits are kept. This is the same reconstruction behavior for images from the true distribution, an example of which can be seen in figure 3 (right) for training images.

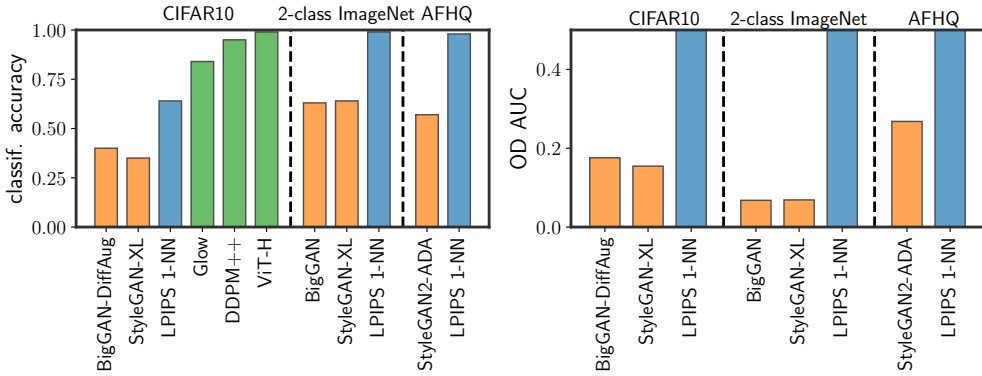

Figure 7: Performance of GANs as generative classifiers (left) and outlier detectors (right). The GANs are outperformed by simple baselines, as well as other generative models.

## 4 Are GANs Good Density Estimators?

Viewing GANs purely as manifold methods potentially ignores two important aspects. First, the prior over the latent codes is completely ignored in all of the tests conducted above. Furthermore, while the GAN is equally far from points in or outside of the distribution, a larger portion of it might pass closer to the training samples than to other samples. An example of this can be seen in figure 6 (left): while both points are equally far from the manifold, more of the manifold passes close to the point in the middle of the spiral ($x_1$), and much less near the other point ($x_2$). Such behavior would explain why GANs are able to generate realistic images from the correct class while still being equally distant from points inside and out of the training distribution. To account for both of these aspects, we evaluate the GANs as density estimators, implicitly taking both the distribution over the latent space and the curvature of the manifold into account.

### 4.1 Adding an Observation Model

When distributions are defined on a low-rank space, it is common to assume an added "observation noise" (e.g. Wu et al. 2016; Kingma & Welling 2013; Tipping & Bishop 1999). In practice, this amounts to the generative model $x = G_\theta(z) +$ noise where the noise is often assumed to be Gaussian with a fixed variance, $\sigma^2$. The density (or likelihood) of a sample under the noisy model is then given by:

$$p_\theta(x) = \int p_\sigma\left(x \mid G_\theta(z)\right) p(z) dz \tag{4}$$

where $p_\sigma\left(x \mid G_\theta(z)\right)$ is the distribution of the observation noise conditioned on the generated sample $G_\theta(z)$. As mentioned by Wu et al. (2016), while the Gaussian observation model is possibly too simplistic, it can used to better understand the distributions learned by GANs. Moreover, as long as this observation is used to compare GANs to each other, then we believe that it does give an indication of which GAN better captures the distribution. Importantly, adding this observation model allows us to take into account situations as described above and in figure 6.

We use the above described observation model in order to estimate the density GANs assign to previously unseen samples, conducting the same experiments as those for the manifold.

In all of our experiments, we use AIS with 500 steps, 4 chains, a Hamiltonian Monte Carlo (HMC) kernel and 10 leap-frogs to estimate the LL of a sample. These hyperparameters were chosen to mitigate the high computational load of running AIS while still extracting accurate estimates of the LL. More experimental details can be found in appendix B.1. The variance of the observation model was chosen to be the mean variance of the reconstruction error of training samples, which is the maximum likelihood estimator for the observation noise. In all cases, the PSNR of the images was high; that is, the variance of the noise

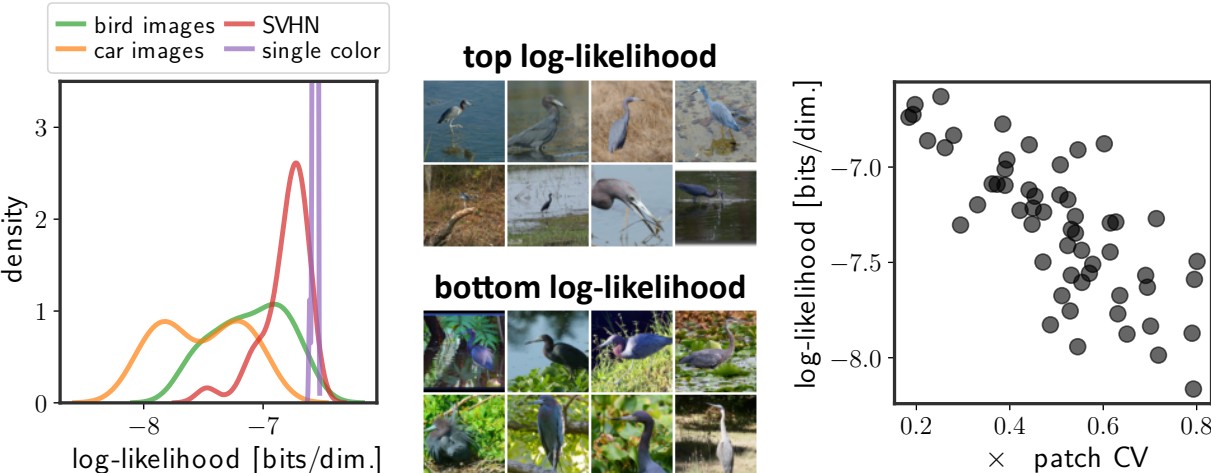

Figure 8: **Left:** log-likelihoods (higher is better) that different image groups attain under a StyleGAN-XL trained only on birds. Images of single color and SVHN images have higher log-likelihoods than images from the domain the GAN was trained on. **Middle:** the top and bottom images according to the log-likelihood. GANs tend to have a bias towards images with larger flat region. **Right:** a scatter plot of the images' log-likelihood against the average coefficient of variation (CV) of $8 \times 8$ patches. The log-likelihood assigned by the GAN is anti-correlated with the average variance of the patches in the image.

assumed was much smaller than the strength of the signal. For more details regarding the variances used, see appendix B.3.

## 4.2 Evaluation of GANs as Density Models

Using the added observation model described above, we calculate the LL of the different GANs on unseen data and compare them to other generative models. This comparison can be found in table 1. GANs assign much lower LL to test samples than other methods, even when their FID is better than the other models.

Figure 7 shows the performance of the GANs as generative classifiers (left) and outlier detectors (right), next to the LPIPS 1NN baseline and other generative models. The classification accuracy of Glow is as reported by Fetaya et al. (2019) and that of DDPM++ as reported by Zimmermann et al. (2021). Once again, the GANs are completely outperformed by the relatively simple 1NN estimator, indicating that the density GANs assign to samples is different than that of the true distribution.

Moreover, GANs tend to assign higher LL to images completely outside their training distribution, as can be seen in figure 8 (left): the StyleGAN-XL bird GAN gave higher LL to SVHN images than to test images.

To further understand the behavior of the GANs as density estimators, we look at which images have the highest and lowest LLs. The images with highest and lowest LLs from the "blue heron" class of ImageNet can be seen in figure 8 (center). Visually it seems that GANs prefer images with larger flat areas and low contrast. Beyond visual inspection, we test whether the LL is effected by local variations in an image by calculating the coefficient of variation for each $8 \times 8$ patch[1] in an image and taking their average. Figure 8 (right) shows that there is a clear anti-correlation between the average local variance within an image and the LL a GAN assigns to images (in this case the "sports car" and "blue heron" classes of ImageNet).

| Model | N=10 | | | N=25 | | | N=50 | | | N=75 | | |
|---|---|---|---|---|---|---|---|---|---|---|---|---|
| | generated | train | svhn | generated | train | svhn | generated | train | svhn | generated | train | svhn |
| **StyleGAN-XL, CIFAR10** | 0.93 | 0.86 | 0.91 | 0.96 | 0.36 | 0.48 | 0.97 | 0.14 | 0.08 | 0.97 | 0.01 | 0.00 |
| **ReACGAN, ImageNet** | 0.97 | 0.14 | 0.82 | 0.96 | 0.00 | 0.45 | 0.94 | 0.00 | 0.07 | 0.97 | 0.00 | 0.00 |
| **StyleGAN-XL, ImageNet** | 0.93 | 0.86 | 0.37 | 0.96 | 0.73 | 0.05 | 0.93 | 0.23 | 0.00 | 0.96 | 0.01 | 0.00 |

Table 2: Typicality tests for different numbers of samples, $N$. In each of the columns, $N$ samples are drawn from either samples generated by the GAN, training images or SVHN images. We run 100 experiments, each time testing whether the drawn samples are part of the typical set $\mathcal{A}_\epsilon^N(p_\theta)$ according to the criterion in equation 5 and report the proportion trails were considered typical.

### 4.3 Typicality of Training Examples

While the samples from the true distribution might have lower LL than other sets of images, this alone is not an indication of a failure of the GAN. A stronger test is that of the *typicality* of a set of images. For the distribution $p_\theta(x)$, we follow Nalisnick et al. (2018) and define the $(\epsilon, N)$-typical set $\mathcal{A}_\epsilon^N(p_\theta)$ of the distribution as all sets of $N$ samples that satisfy:

$$\left| \frac{1}{N} \sum_{i=1}^{N} \log p_\theta(x_i) + H(p_\theta) \right| \leq \epsilon \tag{5}$$

where $H(p_\theta)$ is the entropy of the distribution $p_\theta(x)$. When $N$ is large, then (Cover 1999) these sets of samples capture most of the mass of the distribution defined by $p_\theta(x)$.

We find the $(\epsilon, N)$-typical set by drawing $N$ i.i.d. samples from the GAN. We then calculate their LL using AIS, in exactly the same procedure as that for the training and SVHN images. The average LL of images generated by the GAN gives us an estimate of the entropy of $p_\theta(x)$: $\hat{H}(p_\theta) = -\frac{1}{N} \sum_{i=1}^{N} \log p_\theta(x)$ where $x_i \sim p_\theta(x)$. We set the value of $\epsilon$ by ensuring that 95% of the time, $N$ samples drawn from the model are part of the $(\epsilon, N)$-typical set (i.e. the bootstrap confidence interval suggested by Nalisnick et al. (2019)).

Using the typicality test suggested by Nalisnick et al. (2019), we find that for the GANs we studied the set of *training samples is not part of the $(\epsilon, N)$-typical set* of the distribution defined by the GAN. This can be seen in table 2 which shows the proportion (out of 100 trials) that a set of $N$ images are part of the $(\epsilon, N)$-typical set of the GAN. We test this typicality for images generated by the GAN, training samples and SVHN images. In all cases, a randomly drawn subset of training images has a very low chance of being part of the $(\epsilon, N)$-typical set of the GAN, sometimes even lower than SVHN images.

## 5 Related Works

**Failures of Generative Models.** This work follows a line of works that show that many modern generative models do not truly capture the underlying data distribution. In particular, Nalisnick et al. (2018) showed that models trained using maximum likelihood estimation (MLE) typically assign higher LL to images that are not part of the training distribution, inciting a range of works dedicated to understanding this phenomenon (e.g. Kirichenko et al. 2020; Nalisnick et al. 2019; Zhang et al. 2021; Caterini & Loaiza-Ganem 2022; Fetaya et al. 2019). As far as we know, only generative models trained using MLE have been investigated in this manner by previous works. This work shows that GANs are susceptible to similar weaknesses.

**Mode Collapse in GANs.** As mentioned in the introduction, it has been shown that earlier versions of GANs do not fully capture the training distribution, which was attributed to "mode collapse" in GANs (Goodfellow 2016; Richardson & Weiss 2018). Arora et al. (2018) have quantified mode collapse by a "birthday paradox" test, in which they found that the support of the GANs that were available at the time is smaller than the size of the training distribution. To a large part, mode collapse seems not to be a problem

---

[1]The coefficient of variation of an $8 \times 8$ patch is the standard deviation of the pixel values in the patch divided by the mean pixel value in the same patch.

in modern GANs which capture the full diversity of the training distribution, as evidenced by excellent FID scores. Instead, our analysis points to a deeper problem than previously investigated. Instead of fitting only small parts of the distribution, our results indicate that GANs misrepresent almost all parts of the distribution.

**Density Estimation with GANs.** A key aspect of our investigation is the use of likelihood estimation and inference using GANs. There have been many works in the past that have suggested *extensions* to GANs in order to allow them to work as classifiers or outlier detectors (e.g. Schlegl et al. 2017; Donahue & Simonyan 2019; Kang et al. 2021; Nitzan et al. 2022). Other works change the definition of GANs in order to train them as density estimators together with their adversarial loss (Lucas et al. 2019; Abbasnejad et al. 2019). All such methods use an additional component besides the GAN generator, such as an encoder, which is distinct from the generative process of the GAN. In contrast, we only utilize the tasks of OD and classification as a way to better analyze how well the GAN has captured the true data distribution. We argue that if GANs accurately capture the true data distribution, their generator *alone* should be enough to achieve near-optimal performance in classification/outlier detection.

**Evaluation of GANs.** There is a vast literature on the evaluation of GANs (e.g. Heusel et al. 2017; Sajjadi et al. 2018; Ravuri & Vinyals 2019; Webster et al. 2019; Naeem et al. 2020; Borji 2022; Ravuri et al. 2023). Previous works predominantly use samples from the GAN in order to evaluate them, which raises some uncertainties because the space of all images is huge. In this work, we used the assumptions made when defining GANs in order to evaluate them, thereby side-stepping the issue of determining whether the learned distribution is correct through the use of samples.

**Theoretical Analysis of GANs** There exists a line of research that is dedicated to theoretically examining which distributions different types of GANs are able to describe (Liang 2021; Uppal et al. 2019; Liu et al. 2017) and to describe the training dynamics for GANs (Arjovsky & Bottou 2017; Uppal et al. 2019). As a result of such works, numerous variations of GANs have been proposed, such as WGANs (Arjovsky et al. 2017), MMD-GANs (Arbel et al. 2018), Sobolev-GANs (Mroueh et al. 2017). In this work, we were primarily focused on analyzing the performance of GANs that achieve state-of-the-art results and can generate images that seem to come from the distribution of training images.

## 6 Discussion

Modern GANs are able to generate incredibly realistic images, that seem to arise from the same distribution as the images used to train the GAN. Moreover, they have excellent performance under modern evaluation metrics. These facts have led to a common belief that GANs capture the true data manifold and the true data distribution.

When analyzing GANs as manifold methods, we find that their manifolds do not fit the training samples. Quantitatively, this failure can be summarized by the classification and outlier detection performance of the GANs when defining the distance from the manifold as an estimate for whether the sample is from the GAN's distribution or not. Even when relaxing the requirement that samples should be part of the GAN manifold does not improve the situation, as is evidenced by the poor generative classification behavior. Worse, we showed how the training examples are not part of the typical set of the GAN.

Even though this work has presented an overall negative view of GANs, they are still extremely powerful data samplers. Indeed, this fact was taken advantage of in domains such as image manipulation, style transfer and data augmentation, to great effect. However, our work shows that a more cautious stance should be taken when attempting to use GANs as priors for the true data distribution.

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

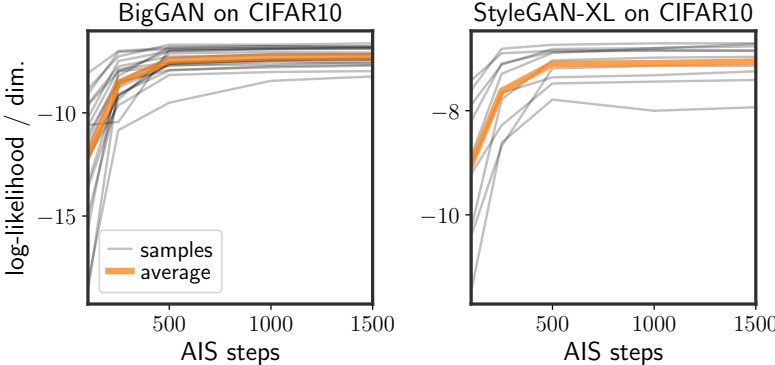

Figure 9: The estimated log-likelihood as a function of AIS steps, for different GANs on CIFAR10 on random test samples. The log-likelihood of most samples converges after 500 steps.

Shengyu Zhao, Zhijian Liu, Ji Lin, Jun-Yan Zhu, and Song Han. Differentiable augmentation for data-efficient gan training. *Advances in Neural Information Processing Systems*, 33:7559–7570, 2020.

Roland S Zimmermann, Lukas Schott, Yang Song, Benjamin A Dunn, and David A Klindt. Score-based generative classifiers. *arXiv preprint arXiv:2110.00473*, 2021.

# A   Brief Explanation of the AIS Algorithm

Let $f(z)$ be a target un-normalized distribution. An AIS chain is defined by an initial distribution $Q_0(z) = q_0(z)/Z_0$ whose normalization coefficient is known, together with $T$ intermediate distributions $Q_1(z), \cdots, Q_T(Z)$ such that $Q_T(z) = q_T(z)/Z_T = f(z)/Z_T$. Each step of the chain further requires an MCMC transition operator $\mathcal{T}_t$ which keeps $Q_t(z)$ invariant, such as Hamiltonian Monte-Carlo (HMC). Beginning with a sample from the initial distribution $z_0 \sim Q_0(z)$ and setting $w_0 = 1$, AIS iteratively carries out the following steps:

$$w_t = w_{t-1} \cdot \frac{q_t(z_{t-1})}{q_{t-1}(z_{t-1})} \qquad z_t \sim \mathcal{T}_t(z|z_{t-1}) \tag{6}$$

The importance weights $w_T$ aggregated during the sampling procedure are an unbiased estimate of the ratio of normalizing coefficients, such that $\mathbb{E}[w_T] = Z_T/Z_0$.

Given a sample $x$, the likelihood can be calculated through AIS by setting $f(z) = p(z)p_\theta(x|z)$ as the target distribution, such that $p_\theta(x)$ is the corresponding normalization constant $\mathcal{Z}_T$. In practice, the likelihood is estimated in log-space to avoid numerical difficulties such as underflows. Calculating the log of the importance weights as described above is straightforward, however Grosse et al. have shown that doing so results in a stochastic lower bound of the log-likelihood. As the number of intermediate steps $T$ increases, this stochastic lower bound becomes tighter and converges to the true log-likelihood.

# B   Implementation Details

## B.1   AIS Details

We follow the implementation of AIS from Wu et al. (publicly available in GitHub), reimplemented in PyTorch. When possible, we used the same settings as Wu et al.:

- The transition operator we used was HMC with 10 leapfrog steps and a Metropolis-Hastings (MH) adjustment. During sampling, the learning rate is initialized to $5 \cdot 10^{-2}$ and adjusted according to a moving average of the MH rejection rate

- During sampling, the intermediate distributions we used were:

$$Q_t(z) \propto p(z) \cdot p_\gamma \left( x | \ G_\theta(z) \right)^{\beta_t} \tag{7}$$

  $\beta_t$ was annealed according to a sigmoidal schedule

**Choice of Number of Steps** The bound on the log-likelihood approximated by AIS becomes tight and accurate only as the number of intermediate distribution and number of chains grows, respectively. However, AIS with many chains and intermediate distributions is incredibly computationally costly. Due to these considerations we use a relatively small number of intermediate distributions, while still ensuring accurate enough results.

Each chain used 500 intermediate steps. This number is in stark contrast to the 10,000 iterations used by Wu et al.. We chose this number by running multiple AIS chains with a differing number of intermediate steps, plotting the estimated log-likelihood as a function of AIS steps, as shown in Figure 9. The estimated log-likelihood typically stabilizes very close to the value reached after 500 iterations. The difference between the converged value and the one after 500 steps is much smaller than the resolution of log-likelihoods we are looking at, so this is a compromise between accuracy and computational cost.

Further justification for this is due to the comparison between AIS and GAN inversion in terms of gradient steps. Because of the leapfrog steps, a single iteration of AIS is similar to 10 gradient steps in GAN inversion. In all of our experiments, $\sim$750 iterations were enough to converge during inversion, well below the 5000 gradient steps used during the AIS procedure.

## B.2 Reconstruction through GAN Inversion

There is a vast literature on the best way to reconstruct test images using GANs, also called *GAN inversion*. In this work we used a simple, albeit rather costly, approach in order to find the best possible reconstruction.

We used an optimization approach towards GAN inversion, using ADAM as the optimizer and a cosine schedule (similar to the scheme used by Sauer et al. in their implementation) for $\sim 500 - 750$ iterations (which we found to usually be many more iterations than required). To find better reconstructions, we sampled $\sim 500$ images from the GAN and initialized the optimizer from the latent code of the image closest to the input image in $\ell_2$ distance. Furthermore, this process was repeated $\sim 4$ times for each image. Using this GAN inversion scheme, we were typically able to invert images generated by the GAN (and frequently images of a single color as well).

Finally, note that for all experiments with the StyleGAN variants, the inversion took place in $\mathcal{Z}$ space, as the generative model is defined in terms of this latent space and not the $\mathcal{W}/\mathcal{W}+$ spaces.

## B.3 Observation Model

To calculate the GANs' density, we assumed an additive Gaussian noise observation model, with the same variance $\sigma^2$ for all samples. This observation model was used by Wu et al. among others. The values of $\sigma^2$ and their respective PSNR can be found in Table 3. As mentioned in the main body of text, the value of $\sigma^2$ used was the variance of the GAN's reconstruction error. This choice of $\sigma^2$ maximizes the likelihood the model gives to training data.

| GAN | Dataset | Value of $\sigma^2$ | PSNR |
|---|---|---|---|
| BigGAN | CIFAR10 | 0.008 | 21.2 |
| StyleGAN-XL | CIFAR10 | 0.018 | 17.4 |
| BigGAN | ImageNet10 | 0.012 | 19.2 |
| StyleGAN-XL | ImageNet10 | 0.021 | 16.8 |
| StyleGAN2-ADA | AFHQ | 0.019 | 17.2 |

Table 3: Table of variances used for $\sigma^2$ throughout the paper and their respective PSNR.

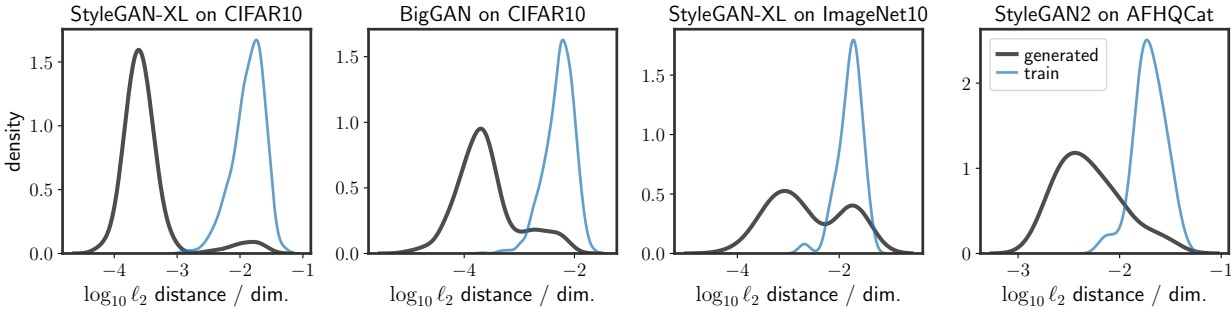

Figure 10: Log reconstruction errors of training vs generated examples for more GANs. In all cases, the distribution of the training examples is different than of the generated examples. The plot for StyleGAN-XL on ImageNet is on a bigger subset of ImageNet classes than in figure 3, showing that this result is broader than the 2 classes discussed in the main body of text.

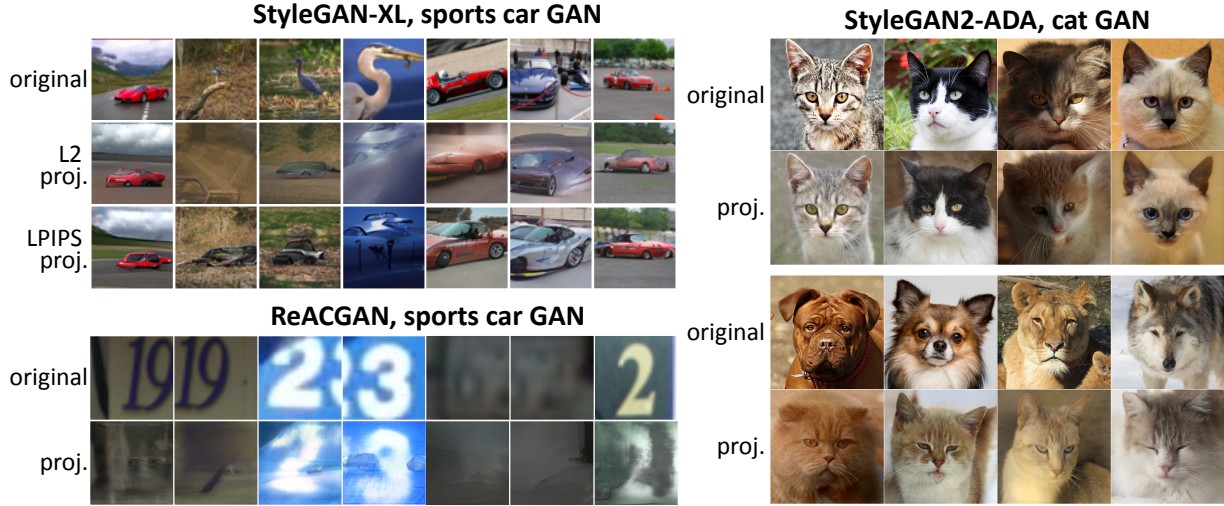

Figure 11: More GAN inversions.

## C    More Results

Figure 10 shows more histograms of the reconstruction errors for training examples vs. images generated by GANs. In all cases, the distribution of the reconstruction errors is different for train and generated examples, indicating that for all GANs examined the training samples are not on the GAN manifold.

Figure 11 shows more examples of reconstructions.

Figure 12 shows LL histograms for different image groups under different GANs.

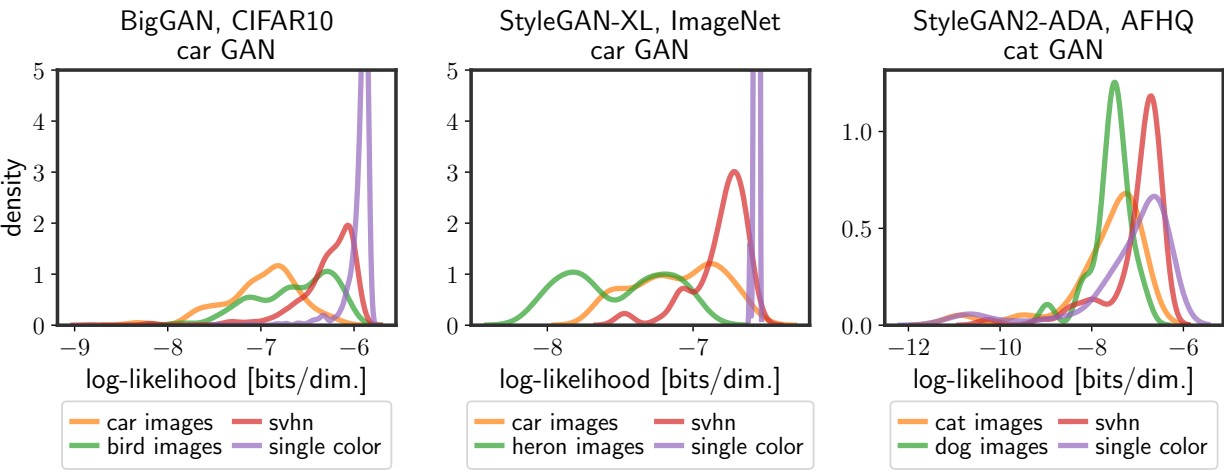

Figure 12: Histograms of LLs for more GANs than shown in the main text.

