# OpenReview forum: "Intriguing Properties of Modern GANs"
_TMLR — Rejected by TMLR_

### Review · Reviewer_iE8j · 2024-03-24

**Summary Of Contributions:**

This paper uncovers some unexpected properties of GANs:

a. Defining the GAN manifold as the image of the generator, the authors show that training images do not exactly lie on the manifold (because they cannot be reconstructed), whereas generated images do lie on the manifold.

b. That the GAN manifolds have poor results when used for outlier detection or classification.

c. By convolving GANs with a small amount of Gaussian noise and then leveraging annealed importance sampling (AIS), the authors do approximate density estimation for a given GAN. They then show that, like likelihood-based models, GANs exhibit the strange pathological behaviour of assigning higher likelihoods to out-of-distribution (OOD) data than they do to in-distribution data.

d. By further leveraging density evaluation, the authors construct typical sets and find that training images are not part of these sets.

Overall, I believe this paper presents some interesting and unexpected results, but the writing should be improved, related literature should be more thoroughly discussed, and some claims should be toned down.

**Audience:**

Yes

**Broader Impact Concerns:**

I have no broader impact concerns.

**Claims And Evidence:**

No

**Requested Changes:**

Address the points raised in each of the weaknesses above.

**Strengths And Weaknesses:**

## Strengths ##

1. I believe the topic of the paper is of interest to the community.

2. Some of the results, such as GANs failing at outlier detection and assigning higher likelihoods to OOD data are unexpected.

3. The sanity checks performed on AIS in the appendix instill confidence that the results are not due to poor density estimates.

## Weaknesses ##

4. I believe that some of the behaviour that the authors call surprising is actually to be expected. Particularly, point (a) in the summary above, which is illustrated in fig 2, is unsurprising: of course generated samples will be closer to the GAN manifold that training samples since generated samples are, by definition, exactly on this manifold! I don't think anyone in the community believes GANs perfectly learn the manifold of training images, but rather that they do a good job at approximating it. I don't think this view is disputed by the results in the paper.

5. The writing of the paper can be improved. In particular:

5a. I found that not enough description was provided to explain how the outlier detection and classification results are obtained from the trained GAN. The only description of how to obtain classifiers is in eq 1, which assumes access to a density model, yet some classification results are presented before it has been explained how density estimation is carried out (fig 3). I assume they were obtained by using a decision stump on some statistic like reconstruction error or log likelihood, but I am unsure, and this should be made explicit.

5b. I would recommend moving some of the content in sec 4.1 to sec 2.2: in the context of GANs, 2.2 makes no sense without the observational noise that is added in sec 4.1, so I did not really understand how the authors could use AIS until 4.1.

6. I don't think the paper does a good job of discussing related work. Very similar pathologies have been observed in likelihood-based models, as pointed out by Nalisnick et al. (2019), which is cited by the authors only in passing. However, there is a large literature looking to further explain the phenomenon, see e.g. [1,2,3] and the citations therein. While I still think that some of the results obtained in this paper are surprising (it is not a priori obvious that the issues afflicting likelihood-based models also affect GANs) the literature exploring the issue for likelihood-based models needs not only to be cited, but also discussed in the context of the empirical results that the authors obtained.

7. In fig 8 the claim is made that, because training data is outside the shaded region, it is outside the typical set, which is claimed to be surprising. However, in all subfigures, most of the black density also lies outside this are, meaning that most generated samples are also outside the typical set? I find this very confusing since by definition, the typical set should contain most generated samples. Could you please clarify?

8. Finally, some minor typos and phrasing issues:

- Please use \citet instead of writing "[author name] et al."

- Page 1: missing period after (Schlegl et al. 2017)

- "We go on to demonstrate that the GAN manifold passes almost the same distance ..." is a weirdly phrased sentence, maybe use language as a manifold containg points rather than passing through them

- Eq 4, while understandable, uses notation you have not introduced yet, please define notation before using it

- "figure 7 shows how the average coefficient of variance in small 8x8 patches of the image is anti-correlated with LL" please add more detail

- Missing comma after eq 5, and missing period after eq 6


[1] Why normalizing flows fail to detect out-of-distribution data, Kirichenko et al., NeurIPS 2020

[2] Understanding failures in out-of-distribution detection with deep generative models, Zhang et al., ICML 2021

[3] Entropic issues in likelihood-based OOD detection, Caterini & Loaiza-Ganem, PMLR 2021

---

> ### Author Response · Authors · 2024-04-09
>
> Thank you for the constructive comments.
>
> Regarding the mentioned problems:
>
> 4. You are of course correct - by design, images generated by the GAN will be on the manifold. We only used these images to verify that our inversion method works: because we know that there exists a latent vector that will be mapped to this image we wanted to verify that our inversion method indeed finds such a latent vector.  In response to the second part, if the GAN were to approximate the image manifold well (in terms of manifold methods), then it has to pass through the training points (or at least close to them). What we showed is that the GAN doesn’t pass through the training points and is actually quite far from them (the distance to the manifold is larger for training images than it is for OOD images such as SVHN, see figure 3).
>
> 5a. Thank you for pointing this out, we will make this clearer in the next version of the manuscript.
>
> **Figure 3**: we used the distance from the manifold as a way to classify points and to judge if they are outliers. To be specific, suppose we observe an image x, we have a cat GAN and a dog GAN. Classification and outlier detection are both carried out by measuring the distance between x and its projection onto the GAN. For classification we measure the distance between x and its projection onto both GANs, classifying according to the GAN whose projection is closest to the original image. For OD we check whether the projection is further than some predefined threshold (the AUC is calculated over a range of possible thresholds).
>
> **Figure 6**: for this figure, classification is carried out according to equation 1, using the likelihood calculated using AIS. OD is similar, where outliers are defined as points with likelihood lower than a predefined threshold and the AUC is over calculated over possible thresholds.
>
> 6. Thank you for the references. We will add them to a more thorough discussion on previous works.
>
> 7. The visualization we chose is perhaps misleading, and we will try to make it clearer in the next version. For typicality, we use the definition in equation 5: a $\left(\epsilon, N\right)$-typical set of the distribution is one that satisfies
> $\left|\frac{1}{N}\sum_{i=1}^N \log p_\theta(x_i) + H\left(p_\theta\right)\right|\le \epsilon$.
> The epsilon is chosen according to a bootstrap estimator such that many subsets of training points are still deemed typical. The entropy of the GANs distribution is the vertical black dashed line in figure 8 (calculated using the AIS log-likelihoods) and the chosen epsilon is the region shaded in gray. Only sets of points whose average log-likelihood falls within this shaded region are typical to the GAN. In other words: only if the average log-likelihood of the data set falls inside the shaded region, it is plausible that it has been generated by the GAN. We then check if the average likelihood for training and SVHN images also fall inside this shaded region and find that they do not - these are depicted by the blue line (training) and red line (SVHN).
>
> 8. Thank you for finding these small mistakes and typos, we will correct them for the next version.

---

> > ### Comment · Reviewer_iE8j · 2024-05-05
> > **Rebuttal acknowledgement**
> >
> > Thank you for your reply and added clarifications, I look forward to seeing the updated version of the manuscript.

---

### Review · Reviewer_GMCs · 2024-04-01

**Summary Of Contributions:**

This paper delves into the intricate aspect of Generative Adversarial Networks (GANs) as manifold-capturing models. While traditional iterations of GANs were susceptible to the mode collapse problem, where the generator would merely memorize training data without generalizing effectively, modern advancements have largely mitigated this issue. Despite these improvements, a widely held assumption persists in the literature that GANs excel at capturing the underlying manifold of training data. However, this paper challenges this assumption through a meticulous reexamination. It unveils a surprising revelation: modern GANs do not accurately capture the training data manifold. Moreover, they even assign higher likelihoods to out-of-distribution data. To substantiate these claims, the paper conducts an extensive empirical analysis leveraging state-of-the-art GAN models.

**Audience:**

Yes

**Broader Impact Concerns:**

I don't see any ethical concerns related to the paper.

**Claims And Evidence:**

Yes

**Requested Changes:**

It would be good to provide some intuition on why the phenomenon described in the paper happens. Here are some questions that I'd like to ask to the authors.
- Are those behaviors of GANs not capturing the training data manifold due to the fact that they do not directly maximize training data likelihood, unlike the other generative models?
- It is quite interesting that GANs prefer images with single colors or images having flatter regions. I also wonder why this is the case.
- What about earlier versions of GANs, especially the ones known to have mode collapse issue. Would they also suffer from similar issues raised in the paper? If so, what makes the modern GANs different from those GANs?

**Strengths And Weaknesses:**

Strengths
- The paper is clearly written and easy to follow.
- The paper brings up an interesting question that the community has overlooked.
- The empirical analysis presented in the paper is impressive both qualitatively and quantitatively. For instance, the log-likelihoods are estimated in a proper way, running AIS for a long number of steps.

Weaknesses
- Even though the empirical results are convincing, the paper does not provide any reasoning or justification on the result; why would GAN fail to capture the training data manifold, or assign higher likelihoods to OOD data?

---

> ### Author Response · Authors · 2024-04-09
>
> Thank you for your review. The questions that you ask are excellent questions and we do not know the answers for them. We hope to be able to answer them in future work but in the meantime we will at least raise them in the next version.
>
> One thing we can say is that the failure cannot be solely attributed to not using likelihood as an objective. Some of the failures we observe with modern GANs (e.g. giving higher likelihood to flat images) also occur in methods that are trained using MLE (e.g. normalizing flow, as shown in Nalisnick et al. 2019).
>
> We also observed similar failures in simple GANs that we trained ourselves (e.g. DCGAN on MNIST) and we will add these to the next version.

---

### Review · Reviewer_6g2U · 2024-04-26

**Summary Of Contributions:**

This paper discusses issues with Generative Adversarial Networks (GAN). The core claim is that the manifolds induced by GAN training do not ``pass-through" the training data manifold but are closer to out-of-distribution data. They support their claims by conducting some empirical studies.

**Audience:**

Yes

**Claims And Evidence:**

No

**Requested Changes:**

The following are the questions that I seek answers for:

1. The whole premise of the paper stands on this statement that the Authors make - This has led many to believe that “GANs capture the true data manifold”. I largely disagree with this. In fact, there is enough evidence to the contrary (Arora et. al.). Given this, I am not sure what the Authors are trying to prove when the community already knows that GANs are mere samplers and NOT manifold learners or density estimators.

2. It is mentioned in the list of contributions that - "The training images are not in the typical set of the GAN". What do Authors mean by the "typical set of the GAN". As far as I know, this is not a standard term in the literature.

3. There is a large set of GAN evaluation metrics that the Authors have missed. I suggest Authors to consider the metrics listed here - Borji, Ali. "Pros and cons of GAN evaluation measures: New developments." Computer Vision and Image Understanding 215 (2022): 103329. Not all metrics compare sample statistics.

4. The paper misses an important reference - Arjovsky, Martin and Bottou, Leon. Towards principled methods for training generative adversarial networks. ICLR 2017. This paper gives a rather different view on the same topic, but in a much more principled manner.

5. In a follow-up paper, Arjovsky et. al, propose remedial measures for the problems of GANs not learning the manifolds. Authors have missed the whole family of WGANs.

6. The argument that "Furthermore, points that are not part of the training distribution are expected to be far from the learned
manifold" is not justified at all. It depends on what is the underlying distance metric used and how are the distances from the manifold measured. On the contrary, the community of adversarial attacks has shown that samples that are `close' too can lead to misclassification. In fact, the whole premise of generating adversarial attacks relies on the fact that it is easy to generate samples `close-enough' to the manifold, yet adversarial.

7. Because of the reasons listed in the previous point, the following statement by the authors also seems unjustified - "If GANs truly behave as believed and shown in figure 1, it should be possible to use them for classification and outlier detection". Samplers (GANs) need not be outlier detectors/classifiers.

8. Inference and Inversion - It is shown that OOD samples can be embedded within the latent space of StyleGAN (Rameen Abdal, Yipeng Qin, and Peter Wonka. Image2stylegan: How to embed images into the stylegan latent space? ICCV 19). How would one express such versatility with GAN's latent spaces?

9. The above-said property is not observed with architectures other than StyleGAN2. Generalization of any property across all GAN models may not be feasible.

10. Overall, I cannot buy Authors' claims unless they are backed by strong theory or strong empirical evidence. In the current form, most of the claims in the paper are either well-known, vague or poorly backed.

**Strengths And Weaknesses:**

Strengths:

1. The paper considers an interesting problem in the field.
2. Most of the experiments considered are interesting.

Weaknesses:

1. The paper makes multiple claims without any theoretical backing.
2. For the most part, the paper is very empirical.
3. It has many vague statements without solid backing.

---

> ### Author Response · Authors · 2024-05-03
>
> 1. We agree that there have been papers that showed that older versions of GANs do not capture the training distribution. However, those papers investigated GANs with much worse performance. For instance, the GANs investigated in the mentioned 2017 Arora et al. paper are DCGANs, which have an FID of 48.29 on CIFAR10 (as reported by Kang et al. 2023 “StudioGAN: a taxonomy and benchmark of GANs for image synthesis”). In contrast, we study modern GANs such as StyleGAN-XL with much better performance, an FID as low as 1.88. This is a vast difference in generative quality. Furthermore, these newer GANs do not suffer from the problems frequently reported in older GANs, such as mode collapse, as evidenced by the good FID, recall and precision reported for them. Regarding the second part of the statement, we are not aware of any previous work that shows that GANs are “mere samplers” and would appreciate some references for this. Regardless, we do not believe that it is well known in the community that GANs do not capture the correct distribution - quoting some research that was published after 2018:
> - “By **learning the distribution of real images via adversarial training**, GANs have advanced image synthesis in recent years.”, from section 1.1 of Zhu et al. 2020 "In-domain gan inversion for real image editing."
> - “Generative Adversarial Network (GAN) is a well known paradigm for **approximating a real data distribution** through an adversarial process.”, from the introduction of Kang et al. 2023 “StudioGAN: a taxonomy and benchmark of GANs for image synthesis”
> - “Generative Adversarial Nets (GANs) have made a dramatic leap in **modeling high dimensional distributions of visual data**.”, from the introduction of Shaham et al. 2019 "Singan: Learning a generative model from a single natural image."
>
>
> 2. The exact definition for the typical set of a distribution is given in section 4.3 of the submitted manuscript and is the same as defined in Nalisnick et al. 2018. The typical test of a distribution is the set of points that are typically expected to be sampled from the distribution, which is distinct from which points have high log-likelihood.
>
>
> 3. We agree that there is a wide range to evaluate GANs. However, the only evaluation methods that are widely reported are IS, FID, precision, recall and similar. All of these methods compare statistics of generated samples to those of training images. We will alter the statement in the paper to make this clear.
>
>
> 4. We will add the suggested reference together with the other mentioned references to a more fleshed out section on related works.
>
>
> 5. As mentioned in the response for the first remark, we are primarily concerned with GANs that perform well in image generation. The family of WGANs are not part of the set of GANs with the highest performance on CIFAR10 or ImageNet (the StyleGAN family, BigGAN-DiffAug, ReACGAN, ProjGAN, ICRGAN, etc.).
>
>
> 6. We agree that the distance metric could potentially have an impact on the definition of “distance from the manifold”, which is why we also considered the LPIPS distance in the submitted manuscript (see figure 3). Even when the much more perceptual distance is used in our experiments, the GANs are considerably outperformed by very simple baselines. More to the point, the images in figure 3 and 4 show that reconstructing training or wrong-class images result in similar behavior - in both the reconstruction captures overall colors but does not capture the object. It should at least be expected that a GAN would reconstruct training images better than images from a completely different distribution.
>
>
> 7. Even if GANs can only generate samples from the correct distribution, it should be able to generate samples that are close to the images it was trained on, but not to images from other classes. Instead, the GANs are NOT able to generate images similar to training images and their quality is as bad as those from the from other classes. On the other hand, the GANs ARE able to sample images that are not part of the original distribution, in particular images of a single color, as seen in figure 4 (left).
>
>
> 8. The mentioned work showed that StyleGAN2 is able to embed images to the W/W+ space of the GAN. In our work, we always search for latent codes in the Z space, as this latent space defines the generative process of the GAN. In StyleGANs, the Z space is mapped to a much larger W space, which is why images could be embedded in the mentioned Abdal et al. paper. However, the distribution of points in W space is unknown and non-trivial, while the Z space has a well-defined prior distribution.
>
>
> 9. As mentioned above, we use the Z space and not the W space, which has the same role across all GANs.

---

### Decision · Action_Editor_TUmJ · 2024-06-12

**Recommendation:** Reject

**Comment:**

While the reviewers thought this paper presents some interesting results, they raised important questions about the claims, the discussion with related literature, and the writing. For instance, two reviewers thought the discussions about "GANs capturing the true data manifold" are to be expected within the community. They found the current revision does not fully address their concerns and needs to better position its claims and discuss how the explanations proposed in that area relate to the authors' observations with GANs. I agree with the reviewers and believe a major revision would be helpful to better convey the points.

In addition, I found the results in Fig. 3 somewhat contradictory to the literature on using GANs for solving image recovery problems. In particular, the works [A, B] and many others have shown that when an image is compressively sampled or corrupted, solving a problem similar to eq. (3) (with an additional linear operator) can provide good recovery, even when the image is not from the training samples. When the linear operator is identity, those approaches become the same as eq. (3). It is not clear why this difference occurs; it may be due to different algorithms used for solving eq. (3), but this is worth considering.

[A] Compressed Sensing using Generative Models, ICML, 2017.
[B] GAN-based projector for faster recovery in compressed sensing with convergence guarantees, ICCV, 2019.

**Audience:**

This paper could be of interest to researchers working on GAN and image generative models.

**Claims And Evidence:**

This paper discusses data manifold perspectives of Generative Adversarial Networks (GAN). The authors empirically show that (1) training images do not exactly lie on the manifold (because they cannot be reconstructed), whereas generated images do lie on the manifold, (2) the GAN manifolds have poor results when used for outlier detection or classification, (3)  the learned density is very far from the data distribution and that GANs tend to assign higher density to out-of-distribution images.

**Resubmission Of Major Revision:**

The authors may consider submitting a major revision at a later time.